# Information-based learning by agents in unbounded state spaces

**Shariq A. Mobin, James A. Arnemann, Friedrich T. Sommer**
Redwood Center for Theoretical Neuroscience
University of California, Berkeley
Berkeley, CA 94720
shariqmobin@berkeley.edu, arnemann@berkeley.edu, fsommer@berkeley.edu

## Abstract

The idea that animals might use information-driven planning to explore an unknown environment and build an internal model of it has been proposed for quite some time. Recent work has demonstrated that agents using this principle can efficiently learn models of probabilistic environments with discrete, bounded state spaces. However, animals and robots are commonly confronted with unbounded environments. To address this more challenging situation, we study information-based learning strategies of agents in unbounded state spaces using non-parametric Bayesian models. Specifically, we demonstrate that the Chinese Restaurant Process (CRP) model is able to solve this problem and that an Empirical Bayes version is able to efficiently explore bounded and unbounded worlds by relying on little prior information.

## 1   Introduction

Learning in animals involves the active gathering of sensor data, presumably selecting those sensor inputs that are most useful for learning a model of the world. Thus, a theoretical framework for the learning in agents, where learning itself is the primary objective, would be essential for making testable predictions for neuroscience and psychology [9, 7], and it would also impact applications such as optimal experimental design and building autonomous robots [3].

It has been proposed that information theory-based objective functions, such as those based on the comparison of learned probability distributions, could guide exploratory behavior in animals and artificial agents [13, 18]. Although reinforcement learning theory has largely advanced in describing action planning in fully or partially observable worlds with a fixed reward function, e.g., [17], the study of planning with internally defined and gradually decreasing reward functions has been rather slow. A few recent studies [20, 11, 12] developed remarkably efficient action policies for learning an internal model of an unknown fully observable world that are driven by maximizing an objective of predicted information gain. Although using somewhat different definitions of information gain, the key insights of these studies are that optimization has to be non-greedy, with a longer time horizon, and that gain in information also translates to efficient reward gathering. However, these models are still quite limited and cannot be applied to agents in more realistic environments. They only work in observable, discrete and bounded state spaces. Here, we relax one of these restrictions and present a model for unbounded, observable discrete state spaces. Using methods from non-parametric Bayesian statistics, specifically the Chinese Restaurant Process (CRP), the resulting agent can efficiently learn the structure of an unknown, unbounded state space. To our knowledge this is the first use of CRPs to address this problem, however, CRPs have been introduced earlier to reinforcement learning for other purposes, such as state clustering [2].

## 2 Model

### 2.1 Mathematical framework for embodied active learning

In this study we follow [12] and use Controlled Markov Chains (CMC) to describe how an agent can interact with its environment in closed, embodied, action-perception loops. A CMC is a Markov Chain with an additional control variable to allow for switching between different transition distributions in each state, e.g. [6]. Put differently, it is a Markov Decision Process (MDP) without the reward function. A CMC is described by a 3-tuple $(\mathscr{S}, \mathscr{A}, \Theta)$ where $\mathscr{S}$ denotes a finite set of states, $\mathscr{A}$ is a finite set of actions the agent can take, and $\Theta$ is a 3-dimensional CMC kernel describing the transition probabilities between states for each action

$$\Theta_{sas'} = p_{s'|s,a} = P(s_{t+1} = s'|s_t = s, a_t = a) \tag{1}$$

Like in [12] we consider the exploration task of the agent to be the formation of an accurate estimate, or *internal model* $\widehat{\Theta}$, of the true CMC kernel, $\Theta$, that describes its world.

### 2.2 Modeling the transition in unbounded state spaces

Let $t$ be the current number of observations of states $S$ and $K_t$ be the number of different states discovered so far. The observed counts are denoted by $C_t := \{\#_1, ..., \#_{K_t}\}$.

Species sampling models have been proposed as generalizations of the Dirichlet process [14], which are interesting for non-parametric Bayesian inference in unbounded state spaces. A species sampling sequence (SSS) describes the distribution of the next observation $S_{t+1}$. It is defined by

$$S_{t+1}|S_1, , S_t \sim \sum_{i=1}^{K_t} p_i(C_t)\delta_{\tilde{S}} + p_{K_{t+1}}(C_t) \tag{2}$$

with $\delta_{\tilde{S}}$ a degenerate probability measure, see [10] for details. In order to define a valid SSS, the sequence $(p_1, p_2, ...)$ must sum to one and be an Exchangeable Partition Probability Function (EPPF). The exchangeability condition requires that the probabilities depend only on the counts $C_t$, not on the order of how the agent sampled the transitions.

Here we consider one of most common EPPF models in the literature, the *Chinese Restaurant Process (CRP) or Polya urn process* [1]. According to the CRP model, the probability of observing a state is

$$p_i(C_t) \quad = \quad \frac{\#_i}{t + \theta} \text{ for } i = 1, ..., K_t \tag{3}$$

$$p_\psi(C_t) \equiv p_{K_{t+1}}(C_t) \quad = \quad \frac{\theta}{t + \theta} \tag{4}$$

where (3) describes revisiting a state and (4) describes the undiscovered probability mass (UPM), i.e., the probability of discovering a new state, which is then labeled $K_{t+1}$. In the following, the set of undiscovered states will be denoted by $\psi$. Using this formalism, the agent must define a separate CRP for each state action pair $s$, $a$. The internal model is then described by

$$\widehat{\Theta}_{sas'} = p_{s'|s,a}(C_t), \tag{5}$$

updated according to (3, 4). The $t$ index in $\widehat{\Theta}_{sas'}$ is suppressed for the sake of notational ease.

Our simplest agent uses a CRP (3, 4) with fixed $\theta$. Further, we will investigate an Empirical Bayes CRP, referred to as EB-CRP, in which the parameter $\theta$ is learned and adjusted from observations online using a maximum likelihood estimate (MLE). This is similar to the approach of [22] but we follow a more straightforward path and derive a MLE of $\theta$ using the EPPF of the CRP and employing an approximation of the harmonic series.

The likelihood of observing a given number of state counts is described by the EPPF of the CRP [8]

$$\pi(C_t; \theta) = \frac{\theta^{K_t}}{\prod_{i=0}^{t-1}(\theta + i)} \prod_{i=1}^{K_t} (\#_i - 1)! \tag{6}$$

Maximizing the log likelihood

$$\frac{d}{d\theta}ln(\pi(C_t;\theta)) = \frac{K_t}{\theta} - \sum_{i=0}^{t-1}\frac{1}{\theta+i} = 0 \tag{7}$$

yields

$$\theta(t) \approx \frac{K_t}{ln(t) + \gamma + \frac{1}{2t} - \frac{1}{12t^2}}, \tag{8}$$

where (8) uses a closed form approximation of the harmonic series in (7) with Euler's Mascheroni constant $\gamma$. In our EB-CRP agent, the parameter $\theta$ is updated after each observation according to (8).

## 2.3 Information-theoretic assessment of learning

Assessing or guiding the progress of the agent in the exploration process can be done by comparing probability distributions. For example, the learning progress should increase the similarity between the internal model, $\widehat{\Theta}$, of the agent and the true model, $\Theta$. A popular measure for comparing distributions of the same dimensions is the KL Divergence, $D_{KL}$. However, in our case, with the size of the underlying state space unknown and states being discovered successively in $\widehat{\Theta}$, models of different sizes have to be compared.

To address this, we apply the following *padding* procedure to the smaller model with fewer discovered states and transitions (Figure 1). If the smaller model, $\widehat{\Theta}$, has $n$ undiscovered state transitions from a known origin state, one splits the UPM uniformly into $n$ equal probabilities (Figure 1a). The resulting padded model is given by

$$\widehat{\Theta}^P_{sas'} = \begin{cases} \frac{\widehat{\Theta}_{sa\psi}}{(|\mathscr{S}_{\Theta_{sa}}|-|\mathscr{S}_{\widehat{\Theta}_{sa}}|)}, & \widehat{\Theta}_{sas'} = 0 & \text{[Figure 1a]} \\ 1/|\mathscr{S}_{\Theta_{sa}}|, & s \notin \mathscr{S}_{\widehat{\Theta}} & \text{[Figure 1b]} \\ \widehat{\Theta}_{sas'}, & \widehat{\Theta}_{sas'} > 0 \end{cases} \tag{9}$$

where $|\mathscr{S}_{\Theta_{sa}}|$ is the number of known states reachable from state $s$ by taking action $a$ in $\Theta$. Further, if there are undiscovered origin states in $\widehat{\Theta}$, one adds such states and a uniform transition kernel to potential target states (Figure 1b).

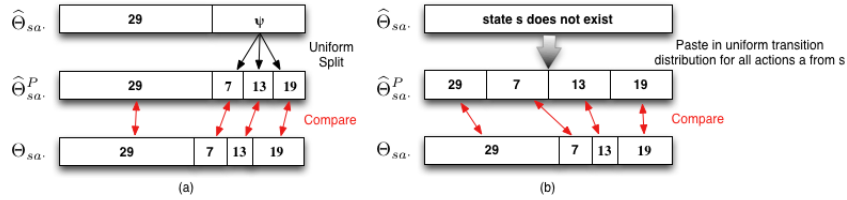

Figure 1: Illustration of the padding procedure for adding unknown states and state transitions in a smaller, less informed model, $\widehat{\Theta}$, of an unbounded environment in order to compare it with a larger, better informed model, $\Theta$. (a) If transitions to target states are missing, we uniformly split the UPM into equal transition probabilities to the missing target states, which are in fact the unknown elements of the set $\psi$. (b) If a state is not discovered yet, we paste this state in with a uniform transition distribution to all target states reachable in the larger model, $\Theta$.

With this type of padding procedure we can define a distance between two unequally sized models,

$$D_{KLP}(\Theta_{sa\cdot}||\widehat{\Theta}_{sa\cdot}) := D_{KL}(\Theta_{sa\cdot}||\widehat{\Theta}^P_{sa\cdot}) := \sum_{s'\in\mathscr{S}_{\Theta_{sa}}} \Theta_{sas'} \log_2\left(\frac{\Theta_{sas'}}{\widehat{\Theta}^P_{sas'}}\right), \tag{10}$$

and use it to extend previous information measures for assessing and guiding explorative learning [12] to unbounded state spaces. First, we define Missing Information,

$$I_M(\Theta||\widehat{\Theta}) := \sum_{s\in\mathscr{S},a\in\mathscr{A}} D_{KLP}(\Theta_{sa\cdot}||\widehat{\Theta}_{sa\cdot}), \tag{11}$$

a quantity an external observer can use for assessing the deficiency of the internal model of the agent with respect to the true model. Second, we define Information Gain,

$$I_G(s, a, s') := I_M(\Theta||\widehat{\Theta}) - I_M(\Theta||\widehat{\Theta}^{s,a \to s'}), \qquad (12)$$

a quantity measuring the improvement between two models, in this case, between the current internal model of the agent, $\widehat{\Theta}$, and an improved one, $\widehat{\Theta}^{s,a \to s'}$, which represents an updated model after observing a new state transition from $s$ to $s'$ under action $a$.

## 2.4 Predicted information gain

Predicted information gain (PIG) as used in [12] is the expected information gain for a given state action pair. To extend the previous formula in [12] to compute this expectation in the non-parametric setting, we again make use of the padding procedure described in the last section

$$
\begin{aligned}
PIG(s, a) \quad &:= \quad E_{s',\Theta|C_t}[I_G(s, a, s')] \\
&= \quad \widehat{\Theta}_{sa\psi} D_{KLP}(\widehat{\Theta}_{sa\cdot}^{s,a \to \eta}||\widehat{\Theta}_{sa\cdot}) + \sum_{s' \in \mathscr{S}_{\widehat{\Theta}_{sa}}} \widehat{\Theta}_{sas'} D_{KL}(\widehat{\Theta}_{sa\cdot}^{s,a \to s'}||\widehat{\Theta}_{sa\cdot}) \quad (13)
\end{aligned}
$$

Here, $D_{KLP}$ handles the case where the agent, during its planning, hypothetically discovers a new target state, $\eta \in \psi$, from the state action pair, $s, a$. There is one small difference in calculating the $D_{KLP}$ from the previous section, which is that in equation (9) $\mathscr{S}_{\Theta_{sa}}$ is replaced by $\mathscr{S}_{\widehat{\Theta}_{sa}^{s,a \to \eta}}$. Thus the RHS of (13) can be computed internally by the agent for action planning as it does not contain the true model, $\Theta$.

## 2.5 Value Iteration

When states of low information gain separate the agent from states of high information gain in the environment, greedy maximization of PIG performs poorly. Thus, like in [12], we employ value iteration using the Bellman equations [4]. We begin at a distant time point ($\tau = 0$) assigning initial values to PIG. Then, we propogate backward in time calculating the expected reward.

$$
\begin{aligned}
Q_0(s, a) \quad &:= \quad PIG(s, a) & (14) \\
Q_{\tau-1}(s, a) \quad &:= \quad PIG(s, a) + \lambda\Big[\widehat{\Theta}_{sa\psi} V_\tau(\psi) + \sum_{s' \in \mathscr{S}_{\widehat{\Theta}_{sa}}} \widehat{\Theta}_{sas'} V_\tau(s')\Big] & (15) \\
V_\tau(s) \quad &:= \quad \max_a Q_\tau(s, a) & (16)
\end{aligned}
$$

With the discount factor, $\lambda$, set to 0.95, one can define how actions are chosen by all our PIG agents

$$a_{PIG} := \underset{a}{argmax}\, Q_{-10}(s, a) \qquad (17)$$

## 3 Experimental Results

Here we describe simulation experiments with our two models, CRP-PIG and EB-CRP-PIG, and compare them with published approaches. The models are tested in environments defined in the literature and also in an unbounded world.

First the agents were tested in a bounded maze environment taken from [12] (Figure 2). The state space in the maze consists of the $|\mathscr{S}| = 36$ rooms. There are $|\mathscr{A}| = 4$ actions that correspond to noisy translations in the four cardinal directions, drawn from a Dirichlet distribution. To make the task of learning harder, 30 transporters are distributed amongst the walls which lead to an absorbing state (state 29 marked by concentric rings in Figure 2). Absorbing states, such as at the bottom of gravity wells, are common in real world environments and pose serious challenges for many exploration algorithms [12].

We compare the learning strategies proposed here, CRP-PIG and EB-CRP-PIG, with the following strategies:

*Random action*: A negative control, representing the minimally directed action policy that any directed action policy should beat.

*Least Taken Action (LTA)*: A well known explorative strategy that simply takes the action it has taken least often in the current state [16].

*Counter-Based Exploration (CB)*: Another explorative strategy from the literature that attempts to induce a uniform sampling across states [21].

*DP-PIG*: The strategy of [12] which applies the same objective function as described here, but is given the size of the state space and is therefore at an advantage. This agent uses a Dirichlet process (DP) with $\alpha$ set to $0.20$, which was found empirically to be optimal for the maze environment.

*Unembodied*: An agent which can choose any action from any state at each time step (hence unembodied) and can therefore attain the highest PIG possible at every sampling step. This strategy represents a positive control.

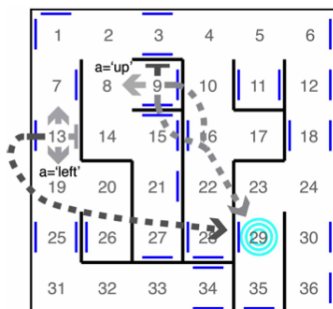

Figure 2: Bounded Maze environment. Two transition distributions, $\Theta_{sa\cdot}$, are depicted, one for ($s$=13, $a$='left') and one for ($s$=9, $a$='up'). Dark versus light gray arrows represent high versus low probabilities. For ($s$=13, $a$='left'), the agent moves with highest probability left into a transporter (blue line), leading it to the absorbing state 29 (blue concentric rings). With smaller probabilities the agent moves up, down or is reflected back to its current state by the wall to the right. The second transition distribution is displayed similarly.

Figure 3 depicts the missing information (11) in the bounded maze for the various learning strategies over 3000 sampling steps averaged over 200 runs. All PIG-based embodied strategies exhibit a faster decrease of missing information with sampling, however, still significantly slower than the unembodied control. In this finite environment the DP-PIG agent with the correct Dirichlet prior (experimentally optimized $\alpha$-parameter) has an advantage over the CRP based agents and reduced the missing information more quickly. However, the new strategies for unbounded state space still outperform the competitor agents from the literature by far. Interestingly, EB-CRP-PIG with continuously adjusted $\theta$ can reduce missing information significantly faster than CRP-PIG with fixed, experimentally optimized $\theta = 0.25$.

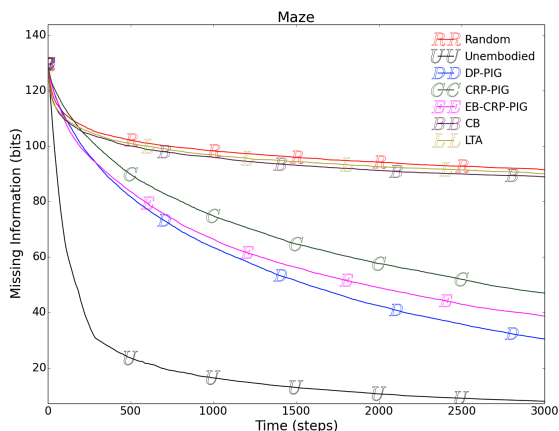

Figure 3: Missing Information vs. Time for EB-CRP-PIG and several other strategies in the bounded maze environment.

To directly assess how efficient learning translates to the ability to harvest reward, we consider the 5-state "Chain" problem [19], shown in Figure 4, a popular benchmark problem. In this environment, agents have two actions available, $a$ and $b$, which cause transitions between the five states. At each time step the agent "slips" and performs the opposite action with probability $p_{slip} = 0.2$. The agent receives a reward of 2 for taking action $b$ in any state and a reward of 0 for taking action $a$ in

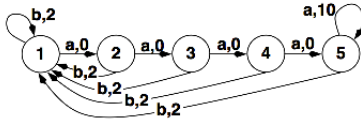 Figure 4: Chain Environment.

every state but the last, in which it receives a reward of 10. The *optimal* policy is to always choose action $a$ to reach the highest reward at the end of the chain, it is used as a positive control for this experiment. We follow the protocol in previous publications and report the cumulative reward in 1000 steps, averaged over 500 runs. Our agent EB-CRP-PIG-R executes the EB-CRP-PIG strategy for S steps, then computes the best reward policy given its internal model and executes it for the remaining 1000-S steps. We found S=120 to be roughly optimal for our agent and display the results of the experiment in Table 1, taking the results of the competitor algorithms directly from the corresponding papers. The competitor algorithms define their own balance between exploitation and exploration, leading to different results.

| Method | Reward |
|---|---|
| RAM-RMAX [5] | 2810 |
| BOSS [2] | 3003 |
| exploit [15] | 3078 |
| Bayesian DP [19] | $3158 \pm 31$ |
| EB-CRP-PIG-R | $3182 \pm 25$ |
| Optimal | $3658 \pm 14$ |

Table 1: Cumulative reward for 1000 steps in the chain environment.

The EB-CRP-PIG-R agent is able to perform the best and significantly outperforms many of the other strategies. This result is remarkable because the EB-CRP-PIG-R agent has no prior knowledge of the state space size, unlike all the competitor models. We also note that our algorithm is extremely efficient computationally, it must approximate the optimal policy only once and then simply execute it. In comparison, the exploit strategy [15] must compute the approximation at each time step. Further, we interpret our competitive edge over BOSS to reflect a more efficient exploration strategy. Specifically, BOSS uses LTA for exploration and Figure 3 indicates that the learning performance of LTA is far worse than the performance of the PIG-based models.

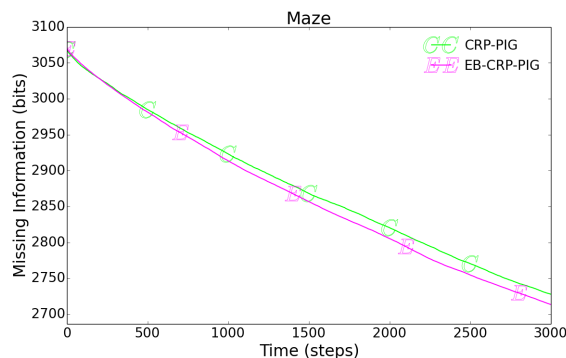

Figure 5: Missing Information vs. Time for EB-CRP-PIG and CRP-PIG in the unbounded maze environment.

Finally, we consider an unbounded maze environment with $|\mathscr{S}|$ being infinite and with multiple absorbing states. Figure 5 shows the decrease of missing information (11) for the two CRP based strategies. Interestingly, like in the bounded maze the Empirical Bayes version reduces the missing information more rapidly than a CRP which has a fixed, but experimentally optimized, parameter value. What is important about this result is that EB-CRP-PIG is not only better but it requires no prior parameter tuning since $\theta$ is adjusted intrinsically. Figure 6 shows how an EB-CRP-PIG and an LTA agent explore the environment over 6000 steps. The missing information for each state is

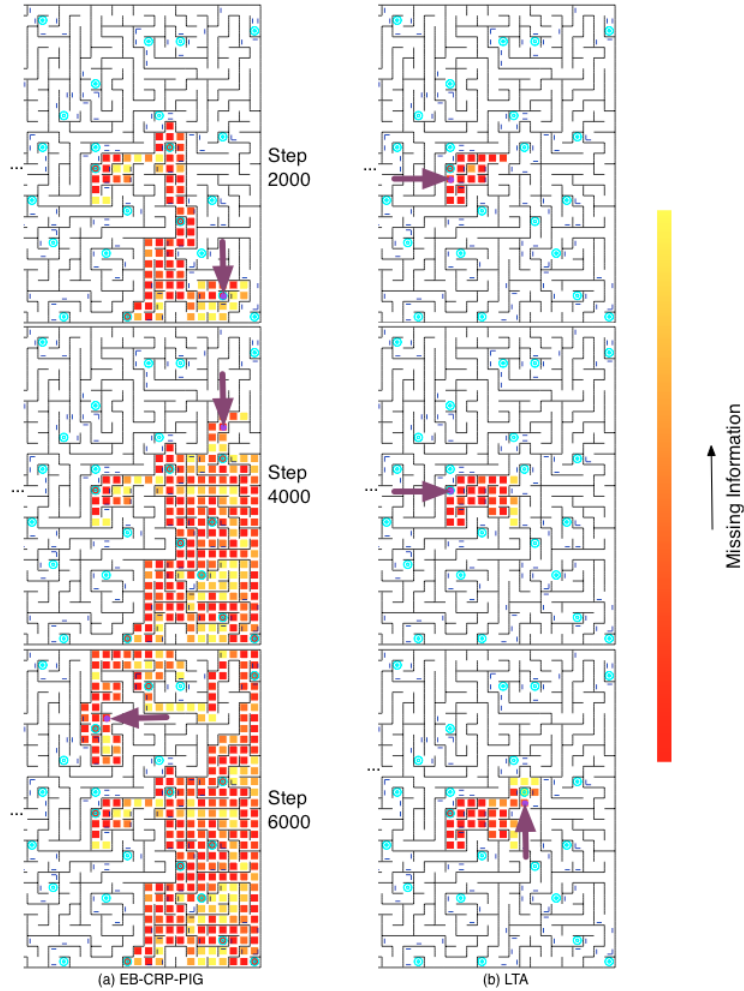

Figure 6: Unbounded Maze environment. Exploration is depicted for two different agents (a) EB-CRP-PIG and (b) LTA, after 2000, 4000, and 6000 exploration steps respectively. Initially all states are white (not depicted), which represent unexplored states. Transporters (blue lines) move the agent to the closest gravity well (small blue concentric rings). The current position of the agent is indicated by the purple arrow.

color coded, light yellow representing high missing information, and red representing low missing information, less than 1 bit. Note that the EB-CRP-PIG agent explores a much bigger area than the LTA agent.

The two agents are also tested in a reward task in the unbounded environment for assessing whether the exploration of EB-CRP-PIG leads to efficient reward acquisition. Specifically, we assign a reward to each state equal to the Euclidian distances from the starting state. Like for the Chain problem before, we create two agents EB-CRP-PIG-R and LTA-R which each run for 1000 total steps, exploring for S=750 steps (defined previously) and then calculating their best reward policy and executing it for the remaining 250 steps. The agents are repositioned to the start state after S steps and the best reward policy is calculated. The simulation results are shown in Table 2. Clearly, the increased coverage of the EB-CRP-PIG agent also results in higher reward acquisition.

| Method | Reward |
|---|---|
| EB-CRP-PIG-R | 1053 |
| LTA-R | 812 |

Table 2: Cumulative reward after 1000 steps in the unbounded maze environment.

# 4 Discussion

To be able to learn environments whose number of states is unknown or even unbounded is crucial for applications in biology, as well as in robotics. Here we presented a principled information-based strategy for an agent to learn a model of an unknown, unbounded environment. Specifically, the proposed model uses the Chinese Restaurant Process (CRP) and a version of predicted information gain (PIG) [12], adjusted for being able to accommodate comparisons of models with different numbers of states.

We evaluated our model in three different environments in order to assess its performance. In the bounded maze environment the new algorithm performed quite similarly to DP-PIG despite being at a disadvantage in terms of prior knowledge. This result suggests that agents exploring environments of unknown size can still develop accurate models of it quite rapidly. Since the new model is based on the CRP, calculating the posterior and sampling from it is easily tractable.

The experiments in a simple bounded reward task, the Chain environment, were equally encouraging. Although the agent was unaware of the size of its environment, it was able to learn the states and their transition probabilities quickly and retrieved a cumulative reward that was competitive with published results. Some of the competitor strategies (exploit [15]) required to recompute the best reward policy for each step. In contrast, EB-CRP-PIG computed the best policy only once, yet, was able to outperform the exploit [15] strategy.

In the unbounded maze environment, EB-CRP-PIG was able to outperform CRP-PIG even though it required no prior parameter tuning. In addition, it covered much more ground during exploration than LTA, one of the few existing competitor models able to function in unbounded environments. Specifically, the EB-CRP-PIG model evenly explored a large number of environmental states. In contrast, LTA, exhaustively explored a much smaller area limited by two nearby absorbing states.

Two caveats need to be mentioned. First, although the computational complexity of the CRP is low, the complexity of the value iteration algorithm scales linearly with the number of states discovered. Thus, tractability of value iteration is an issue in EB-CRP-PIG. A possible remedy to this problem would be to only calculate value iteration for states that are reachable from the current state in the calculated time horizon. Second, the described padding procedure implicitly sets a balance between seeking to discover new state transitions versus sampling from known ones. For different goals or environments this balance may not be optimal, a future investigation of alternatives for comparing models of different sizes would be very interesting.

All told, the proposed novel models overcome a major limitation of information-based learning methods, the assumption of a bounded state space of known size. Since the new models are based on the CRP, sampling is quite tractable. Interestingly, by applying Empirical Bayes for continuously updating the parameter of the CRP, we are able to build agents that can explore bounded or un-bounded environments with very little prior information. For describing learning in animals, models that easily adapt to diverse environments could be crucial. Of course, other restrictions in these models still need to be addressed, in particular, the limitation to discrete and fully observable state spaces. For example, the need to act in continuous state spaces is obviously crucial for animals and robots. Further, recent literature [7] supports that information-based learning in partially observable state spaces, like POMDPs [17], will be important to address applications in neuroscience.

# 5 Acknowledgements

JAA was funded by NSF grant IIS-1111765. FTS was supported by the Director, Office of Science, Office of Advanced Scientific Computing Research, Applied Mathematics program of the U.S. Department of Energy under Contract No. DE-AC02-05CH11231. The authors thank Bruno Olshausen, Tamara Broderick, and the members of the Redwood Center for Theoretical Neuroscience for their valuable input.

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
