[Reviews · NeurIPS 2014]

Submitted by Assigned_Reviewer_11

In their paper “Information-based learning by agents in unbounded state spaces” the authors extend a previous model of information-based exploration as described in reference [11] to unbounded state spaces by introducing a Chinese restaurant process to model transition probabilities.

Previous studies have used the Chinese restaurant process for reinforcement learning—for example, reference [2] cited by the authors. It would therefore be good if the authors could clarify the differences to previous studies that have used Chinese restaurant processes in reinforcement learning to clarify originality.

L 130: verb is missing in the second part of the sentence

To compute the information gain the authors need to compute relative entropies between the true state transition distribution and the estimated state transition distribution. As explained by the authors, these two distributions initially have different dimensionality. To make the distributions commensurable they expand the dimensionality of the estimated distribution to include states that have not been observed to date by uniform distributions. However, this seems to defeat the purpose of the paper to some extent, since the true cardinality of all random variables has to be known in equation (9) and all subsequent equations (10)-(13). This is a crucial question to clarify.

L 160
Question wrt equation (12)
The last \hat{\theta}_sa is the updated model? In that case shouldn’t it be written as \hat{\theta}^{a,s\rightarrow s^*} as in equation (6) in reference [11]?

The information gain measure the authors use in equation (13) is somewhat unconventional. Therefore, it would be important to see how it performs in the most simple exploration problem—that is the bandit problem. This should be even simpler to simulate than the experiments they present. For bandit problems there are well established bounds on how efficient different exploration methods are and the authors could compare against established schemes like UCB or Thompson sampling.
Summary: In summary, the paper proposes an interesting way to deal with environments with unbounded states. However, it seems that the number of states has to be known beforehand, which somewhat defeats the purpose of the paper. Moreover, the original contribution of the paper should be clarified.

Submitted by Assigned_Reviewer_26

The paper proposes a learning model for unbounded, observable discrete state spaces. The author(s) extended a work in [11] to a nonparametric Bayesian framework and introduced some tricks so that it can assess the similarity between models of different sizes.
The experimental results show that the proposed model performs better than the competitors.

[quality]
The proposed model is a natural extension of the existing one to a nonparametric Bayesian framework.

[clarity]
The paper is well written and rather easy to follow.

[originality]
This work is a combination of an existing model and Bayesian nonparametrics.

[significance]
The method may be useful if a user has a problem that satisfies the discussed condition.
Summary: Although the proposed model may be useful, it is a natural extension of an existing model to Bayesian nonparametrics and has rather less novelty.

Submitted by Assigned_Reviewer_31

The paper „Information-based learning by agents in unbounded state space“ presents an algorithm that guides a learning agent to explore a new probabilistic environment with a possibly infinite number of states. The guidance principle ist the maximization of information of the learned model. The authors use a Chinese restaurant process in order to generalize the underlying Dirichlet process for an unbounded number of states.

The formalism of the paper is not easy to follow, even more so as the nomenclature of the central variables is not strictly kept throughout the paper. The subscript and the functional argument of p has different meanings in (2) as opposed to (3) and (4). And it is not wise to omit the explicit dependency on s and a in (5). The value iteration and the time horizon should be described in this very paper and the reader should not only be refered to [11].

The extension of the described algorithm to an unbounded number of states is straight-forward but original and the question of learning through autonomous exploration in MDPs is an important one to the NIPS community.

Some typos:
P2, 104: ”…number of state counts IS …”
P3, 130: “To make … possible.” is not an English sentence
Summary: The paper „Information-based learning by agents in unbounded state space“ presents an algorithm that guides a learning agent to explore a new probabilistic environment with a possibly infinite number of states. The formalism of the paper is not easy to follow, even more so as the nomenclature of the central variables is not strictly kept throughout the paper.
Author Feedback
Author rebuttal: * Response to all Reviewers *
We appreciate the thorough reading of our work and would like to thank all of our reviewers for the constructive feedback and opportunity to respond.

* Responses to assigned Reviewer 11 *
// Lack of discussion about CRP
In reference [2] the CRP is used to find clusters of states with identical state transition probabilities. In contrast, we use the CRP to describe the state transition probabilities themselves. We did not find references beyond [2] that use CRPs in RL. We will add this discussion in a revised version.

// The number of states has to be known beforehand
This impression is not true, but we recognize that we did not explain equations (9)-(13) and the associated diagram well enough. The diagram below equation (9) describes how an ***external observer*** measures the missing information of the agent. It does not describe how the ***agent*** itself measures the information gained for taking certain actions. The agent does not need access to the true state transition distribution but instead uses the hypothetical state transition distribution and then follows equations (9)-(13). The agent therefore does not require knowledge of the actual number of states beforehand.

// Error in Equation (12)
Correct, thank you for pointing this out.

// Information gain measure unconventional and relevance to the Bandit Problem
It is informative to consider PIG in the context of the bandit problem. We do not think that PIG, combined with a reward objective, would be competitive. This is because PIG assesses the global information gain, the gathering of information for all actions. In contrast, a good solution to the bandit problem gathers information for specific actions which have potentially high reward and ignores others.

* Responses to assigned Reviewer 26 *
// The proposed model may be useful but is not very novel
We acknowledge that our model which combines information-based learning and non-parametric bayes is not in itself a major innovation in machine learning. However, we think our work can contribute significantly to the fields of neuroscience and psychology to help understand exploratory behavior. Specifically, our model provides a theoretical framework for how embodied agents can learn the structure of unknown environments in a self-driven manner.

* Responses to assigned Reviewer 31 *
// The formalism of the paper is not easy to follow (Equations (2)-(4))
We recognize that the notation in equations (2)-(4) is confusing and we will fix the mentioned problems.

// It is not wise to omit the explicit dependency on s and a in equation (5)
We agree with this assessment and will change equation (5) accordingly.

// The value iteration and the time horizon should be described in this very paper
We agree that our discussion of value iteration and the time horizon is incomplete. We will address these omissions to make the paper stand alone.